# Health Inequalities in Primary Care: A Comparative Analysis of Climate Change-Induced Expansion of Waterborne and Vector-Borne Diseases in the SADC Region

**DOI:** 10.3390/ijerph22081242

**Published:** 2025-08-08

**Authors:** Charles Musarurwa, Jane M. Kaifa, Mildred Ziweya, Annah Moyo, Wilfred Lunga, Olivia Kunguma

**Affiliations:** 1Department of Science, Technology and Design Education, Faculty of Education, Midlands State University, Gweru P. B. 9054, Zimbabwe; kaifaj@staff.msu.ac.zw (J.M.K.); moyoan@staff.msu.ac.zw (A.M.); 2Department of Accounting and Business Education, Hillside Teachers College, Hillside, Bulawayo P. B. 2, Zimbabwe; midy1978@gmail.com; 3Developmental Capable Ethical State, Human Sciences Research Council, Pretoria 0001, South Africa; wlunga@hsrc.ac.za; 4Disaster Management Training Education Centre (DIMTEC), Faculty of Natural and Agricultural Sciences, University of Free State, Bloemfontein 9301, South Africa; kungumao@ufs.ac.za; 5African Centre for Disaster Studies (ACDS), Faculty of Natural and Agricultural Sciences, Northwest University, Potchefstroom 2531, South Africa

**Keywords:** adaptive capacity, vulnerabilities, waterborne diseases, high-risk areas, SADC region, climate change health

## Abstract

Climate change has magnified health disparities across the Southern African Development Community (SADC) region by destabilizing the critical natural systems, which include water security, food production, and disease ecology. The IPCC (2007) underscores the disproportionate impact on low-income populations characterized by limited adaptive capacity, exacerbating existing vulnerabilities. Rising temperatures, erratic precipitation patterns, and increased frequency of extreme weather events ranging from prolonged droughts to catastrophic floods have created favourable conditions for the spread of waterborne diseases such as cholera, dysentery, and typhoid, as well as the expansion of vector-borne diseases zone also characterized by warmer and wetter conditions where diseases like malaria thrives. This study employed a comparative analysis of climate and health data across Malawi, Zimbabwe, Mozambique, and South Africa examining the interplay between climatic shifts and disease patterns. Through reviews of national surveillance reports, adaptation policies, and outbreak records, the analysis reveals the existence of critical gaps in preparedness and response. Zimbabwe’s Matabeleland region experienced a doubling of diarrheal diseases in 2019 due to drought-driven water shortages, forcing communities to rely on unsafe alternatives. Mozambique faced a similar crisis following Cyclone Idai in 2019, where floodwaters precipitated a threefold surge in cholera cases, predominantly affecting children under five. In Malawi, Cyclone Ana’s catastrophic flooding in 2022 contaminated water sources, leading to a devastating cholera outbreak that claimed over 1200 lives. Meanwhile, in South Africa, inadequate sanitation in KwaZulu-Natal’s informal settlements amplified cholera transmission during the 2023 rainy season. Malaria incidence has also risen in these regions, with warmer temperatures extending the geographic range of Anopheles mosquitoes and lengthening the transmission seasons. The findings underscore an urgent need for integrated, multisectoral interventions. Strengthening disease surveillance systems to incorporate climate data could enhance early warning capabilities, while national adaptation plans must prioritize health resilience by bridging gaps between water, agriculture, and infrastructure policies. Community-level interventions, such as water purification programs and targeted vector control, are essential to reduce outbreaks in high-risk areas. Beyond these findings, there is a critical need to invest in longitudinal research so as to elucidate the causal pathways between climate change and disease burden, particularly for understudied linkages like malaria expansion and urbanization. Without coordinated action, climate-related health inequalities will continue to widen, leaving marginalized populations increasingly vulnerable to preventable diseases. The SADC region must adopt evidence-based, equity-centred strategies to mitigate these growing threats and safeguard public health in a warming world.

## 1. Introduction

Climate change has transcended its traditional framing as merely an environmental or economic issue, emerging as a defining determinant of global public health. It destabilizes the core pillars that sustain human well-being, that is, safe drinking water, clean air, nutritious food, and secure shelter [1]. The health consequences of climate change manifest through a spectrum of direct and indirect pathways. Direct impacts are observable in the increasing number of illnesses and deaths resulting from extreme weather events such as heatwaves, floods, and storms [2]. At the same time, indirect effects emerge from ecologically and socially mediated mechanisms, which include shifting patterns of vector-borne diseases, the contamination of water sources, the disruption of agricultural systems, the displacement of populations, heightened resource-based conflict, and the collapse of essential infrastructure [3,4].

Efforts to quantify these multifaceted risks have been advanced by the World Health Organization through standardized comparative risk assessment models, enabling the estimation of disease burdens attributable to climate change [5]. Applying these frameworks, researchers found that by the year 2000, compared to a 1961–1990 baseline, climate change had already resulted in approximately 160,000 additional deaths annually and 5.5 million disability-adjusted life year (DALY) losses, primarily due to malaria, diarrheal illnesses, malnutrition, and extreme weather events [5]. Updated projections indicate that these numbers are increasing steadily, with over 250,000 annual deaths predicted between 2030 and 2050 if current levels of mitigation persist [1]. This analysis draws upon conceptual foundations from [6] an eco-epidemiological model and [7] coupled socio-environmental systems approach to offer a structured understanding of climate-related health impacts. The first domain, direct health effects, includes rising rates of thermal stress-related mortality, especially in urban areas and among those exposed to outdoor heat, such as labourers and the elderly [8]. Injuries and fatalities from climate-related disasters such as Cyclone Idai in Mozambique (2019) and the devastating 2022 floods in Pakistan illustrate the growing severity of these hazards [9]. Indirect ecological pathways are equally concerning. The geographic spread of disease vectors like Aedes and Anopheles mosquitoes has expanded due to warming temperatures, increasing the prevalence of diseases such as malaria. Notably, highland regions in East Africa have experienced a 20% rise in malaria incidence since 2000 [10]. Simultaneously, climate-related changes such as drought and ocean acidification are jeopardizing food production and fisheries, exacerbating nutritional deficiencies and contributing to stunting in children [11].

Social and systemic factors act as powerful amplifiers of these risks. Climate-induced displacement is projected to affect as many as 1.2 billion people by 2050, placing immense strain on urban slums where infectious diseases flourish [12]. Furthermore, health systems already weakened by concurrent crises, such as the COVID-19 pandemic, face reduced adaptive capacity to respond effectively to new climate-related challenges [4,13].

Most critically, the global health burden induced by climate change is marked by deep inequity. Low-income countries, which have contributed less than 10% of cumulative greenhouse gas emissions, disproportionately suffer more than 75% of climate-related health impacts (WHO 2021). Thus, the urgent need for comprehensive mitigation and adaptation strategies, ranging from early warning systems for heatwaves to climate-resilient urban planning, cannot be overstated. These interventions are essential for safeguarding health and ensuring equitable resilience in the face of a rapidly changing climate [2].

### 1.1. Research Questions

The following research questions were used to guide the study:How is climate change influencing the geographic and temporal spread of waterborne and vector-borne diseases in SADC?How do these diseases disproportionately affect marginalized populations, exacerbating health inequalities?What gaps exist in primary care systems’ capacity to address climate-sensitive diseases?

### 1.2. Problem Statement

The Southern African Development Community (SADC) region is grappling with a pressing public health concern, as the incidence of vector-borne diseases continues to escalate, posing a significant threat to the well-being of its citizens. Climate change is exacerbating this issue, as rising temperatures, changing precipitation patterns, and increased frequency of extreme events create ideal breeding conditions for vectors, thereby expanding the geographic range of disease transmission. This, in turn, puts a strain on the region’s health systems, particularly in rural and underserved areas, where access to health care services is already limited. The impact is multifaceted, resulting in increased morbidity and mortality, as well as significant economic losses due to health care costs, lost productivity, and impact on tourism and trade. The region’s vulnerability to climate–health-related risks is further complicated by socioeconomic and infrastructural challenges, underscoring the need for a comprehensive and integrated approach to address the health implications of climate change in the SADC region.

## 2. Theoretical Framework

The theoretical framing is anchored in McMichael’s eco-social model of climate and health [3], which emphasizes the interaction between ecological disruption and social vulnerability in shaping disease burden. The model emphasizes the importance of understanding the dynamic interaction between climate, ecosystems, and human health, and highlights the need for a holistic approach to addressing the impacts of climate change. The eco-social model consists of several key components, including the natural environment, human systems, and health outcomes. The natural environment includes the physical and biological systems that support life on earth such as the atmosphere, oceans, and ecosystems. Human systems include the social, economic, and cultural factors that influence human health and well-being.

This model recognizes that climate change is a multifaceted issue that affects human health through various pathways, including direct and indirect, immediate and delayed, and localized and diffuse impacts [14]. Direct impacts include mortality and morbidity due to extreme weather events such as heatwaves, floods, and storms. Indirect impacts include changes in the distribution and transmission of vector-borne diseases, such as malaria and dengue fever, and changes in water and food security.

The study applied the eco-social model by examining several pathways, which included the impact of climate change on ecosystems and the natural environment in the SADC region, the impact of climate change on the social and economic determinants of health in the SADC region, the impact of climate change on the incidence and prevalence of water borne and vector-borne diseases in the SADC region, and finally the development of effective adaptation strategies to reduce the risks associated with climate change.

McMichael’s eco-social model contributed to a deeper understanding of the complex relationship between climate change, ecosystems, and human health and informed the development of effective policies and interventions to address the health impacts of climate change in the SADC region. This is further supported by the One Health approach [15], which highlights the interdependence of human, animal, and environmental health critical in vector-borne disease dynamics and food–water security. To capture the structural limitations of PHC systems, the study draws on the Health System Resilience Framework [16], which outlines key capacities including absorptive, adaptive, and transformative resilience necessary for effective response to climate shocks.

## 3. Review of Relevant Literature

An analysis of relevant documents reveals that there is a correlation between climate change indicators and the increased occurrence of various diseases across the sampled four SADC states of Malawi, Zimbabwe, Mozambique, and South Africa [1,2,3,4]. Studies indicate that rising temperatures and altered precipitation patterns contribute to the spread of malaria, cholera, and diarrheal diseases in these regions [2,5,6]. Additionally, extreme weather events, such as floods and droughts, exacerbate waterborne and vector-borne disease risks [3,7]. Table 1 shows changes in two key climatic elements (temperature and rainfall) from 2015 to 2023. Across the four countries, temperatures increased by an average of 1.25 °C [1,2], while rainfall patterns became more erratic, characterized by intensified cyclones in Mozambique [3,4] and prolonged droughts in Zimbabwe and Malawi [5,6]. South Africa (with temperate/Mediterranean climates) faced intensified heatwaves in summers and flooding in winter rainfall zones (e.g., Cape Province) [7,8]. KwaZulu-Natal (KZN) experienced frequent floods due to its exposure to the warm Agulhas Current [9,10]. Malawi (near the Equator and Indian Ocean) saw increased flood frequency [11,12]. These trends align with IPCC AR6 regional projections [13] and thet WHO climate–health vulnerability assessments [14].

Thus, as shown in Table 1, a correlation between average temperature increases, rainfall variability, and the existence of extreme weather events can be deduced. The introduction of a third variable, which is disease, also reveals the notion that there is also a marked increase in the occurrence of waterborne and vector-borne diseases. This correlation is shown in Table 2.

Malaria, cholera, and dengue fever are climate-sensitive diseases that have increased across all four SADC countries, with Mozambique experiencing the highest cholera outbreaks following floods. Malaria is a vector-borne disease transmitted by Anopheles mosquitoes, which breed in warm, humid environments where stagnant water is present. Rising temperatures, combined with increased rainfall or water stagnation, create optimal conditions for mosquito proliferation, thereby increasing the risk of malaria transmission. Climate variability, particularly heatwaves, altered precipitation patterns, and extreme weather events, has been shown to influence the geographic and seasonal spread of malaria, especially in tropical and subtropical regions [1,2].

Although malaria is most prevalent in Sub-Saharan Africa, it also affects countries in other tropical zones, including the nine nations bordering the Amazon Basin in South America, eight countries in Central America and the Caribbean, and parts of Southeast Asia and the Eastern Mediterranean [3]. Post-disaster environments are particularly vulnerable to vector-borne disease outbreaks due to disrupted sanitation systems, standing water, and compromised health services. For instance, following Cyclone Idai, districts in central Mozambique experienced significant outbreaks of dengue fever, linked to rising post-disaster temperatures and widespread water pooling—conditions ideal for Aedes mosquito breeding [4]. These patterns underscore the critical need for integrating climate surveillance into vector control and public health preparedness strategies. Climate change-induced hazards and risks do have a massive impact on Primary Health Care (PHC) systems in each of the four countries, as shown in Table 3.

Extreme weather and disease outbreaks have strained Primary Health Care (PHC) systems, particularly in Mozambique, where flooding destroyed critical infrastructure. Drug stockouts and overcrowding compromised service delivery, while health workforce gaps worsened rural care access [1]. Data indicate that climate-induced disease outbreaks have intensified pressure on PHC systems across Malawi, Zimbabwe, Mozambique, and South Africa [2,3]. Increased frequency of floods, cyclones, and droughts has disrupted services, damaged facilities, and restricted access to care [4,5]. In Malawi, recurrent cholera and malaria outbreaks following floods overwhelmed understaffed rural clinics with inadequate medical supplies [6,7]. Mozambique’s PHC system suffered severe damage during Cyclone Idai (2019), which destroyed 90+ health facilities and disrupted immunization programs [8]. Zimbabwe experienced typhoid and diarrheal disease outbreaks during droughts, exacerbating water shortages and undermining preventive care [9]. In South Africa, the 2022 KwaZulu-Natal floods forced PHC clinic closures, delaying maternal and child health services [10].

These trends reveal systemic gaps in climate-resilient health planning, including weak disease surveillance, fragile infrastructure, and insufficient emergency preparedness [11,12]. Strengthening climate-adaptive PHC systems and early warning mechanisms is critical to maintaining care during climate shocks [13,14].

## 4. Methodology

This study adopted a multisite case study design to explore the post-disaster contexts of Malawi, Zimbabwe, Mozambique, and South Africa, all of which are increasingly vulnerable to the health consequences of climate change-induced disasters. The research employed a mixed-methods approach, integrating a qualitative comparison of climate variability based on rainfall and temperature indicators—with a quantitative epidemiological analysis of health outcomes and the performance of Primary Health Care (PHC) systems. This was triangulated with qualitative policy analysis to deepen insight into climate–health dynamics in the region, consistent with recent methodological frameworks in climate–health scholarship [1,2].

Document analysis included national health reports, meteorological data (e.g., temperature and precipitation records from 2015 to 2023), disease surveillance databases, and outbreak records from national ministries of health and global repositories. Particular attention was paid to climate-sensitive diseases such as cholera, malaria, and diarrheal infections, with temporal–spatial patterns in disease occurrence examined alongside qualitative narratives explaining climate-related health shocks. This approach reflects best practice in integrated climate–health surveillance and is aligned with the growing literature on planetary health and public health system resilience in Sub-Saharan Africa [3,4].

A case-based comparative method enabled detailed examination of climate–health events across the four countries. For example, the 2022 cholera outbreak in Malawi following Cyclone Ana was traced through causal chains linking extreme rainfall, WASH infrastructure failure, and overwhelmed PHC facilities paralleling findings in recent analyses of extreme weather–disease linkages [5]. In South Africa, informal settlement case studies in KwaZulu-Natal provided insight into how inadequate sanitation and uneven PHC access worsened disease transmission during the 2023 flood events, supporting previous observations of infrastructure vulnerability in urban marginal contexts [6].

The use of integrated climate–health surveillance as an analytic lens revealed critical systemic gaps, including limited early warning capabilities, weak community outreach, and fragmented multisectoral coordination. These findings are consistent with the World Health Organization’s call for health-informed climate adaptation and the need for equity-centred, data-driven PHC systems [7]. Ultimately, the study underscores how climate change exacerbates existing health disparities, reinforcing the imperative for PHC systems to become central actors in both mitigation and adaptation strategies.

## 5. Findings and Discussion

### 5.1. Climate Change and the Shifting Disease Burden in the SADC Region: Implications for Primary Health Care Systems

Climate change is profoundly reshaping the epidemiological landscape of the Southern African Development Community (SADC) region, intensifying health risks and exacerbating existing public health inequities. The region is experiencing rising average temperatures, more frequent and severe droughts, intense flooding, cyclones, and shifting precipitation patterns, all of which alter the distribution and incidence of communicable diseases [26]. These climate-driven environmental changes directly and indirectly impact the burden of disease, particularly waterborne and vector-borne illnesses, while placing unprecedented strain on fragile PHC systems, especially in rural and informal urban settings.

Health data across Malawi, Zimbabwe, Mozambique, and South Africa reveals the interplay between climatic shifts and disease patterns including various sources (see Table 4).

In Malawi, the Nsanje District—the southernmost and flood-prone area—is a clear example. Cyclone Ana in 2022 submerged villages like Bangula and displaced over 200,000 people. The floods damaged boreholes and latrines, contaminating water sources and triggering a severe cholera outbreak [27]. The Bangula Rural Health Centre, serving a population of nearly 50,000, reported drug stockouts and lacked rehydration kits, resulting in preventable fatalities [28]. Health workers were unable to reach remote communities due to destroyed roads, severely limiting surveillance and timely treatment.

In Mozambique, the Buzi District in Sofala Province experienced catastrophic flooding after Cyclone Idai in 2019. Health posts in communities like Guara Guara and Estaquinha were submerged, forcing residents to rely on overcrowded makeshift clinics. Cholera cases tripled in the weeks following the disaster, particularly affecting children under five and pregnant women [29]. The Buzi Rural Hospital struggled with staff shortages and lacked isolation facilities. Moreover, standing floodwaters created breeding grounds for Anopheles funestus, increasing malaria incidence, a challenge further exacerbated by disrupted indoor residual spraying campaigns.

Zimbabwe’s Gwanda District, located in drought-stricken Matabeleland South, has seen recurrent outbreaks of diarrheal diseases due to water scarcity. In villages like Ntalale and Guyu, residents travel long distances to unprotected wells shared with livestock. Beitbridge Rural, a border town community, faced significant cholera risk due to cross-border movement, poor sanitation, and limited water supply [30]. In 2019, the local PHC clinic in Ntalale recorded a doubling of diarrheal cases among children. With erratic rainfall patterns and declining groundwater levels, households resort to unsafe water storage practices, creating conditions conducive to the spread of waterborne infections [31,32]. These facilities often lack access to running water, further impeding infection control. Chikukwa, Chiramba, and Dombe communities in Chimanimani District were isolated due to landslides and flooding. These rural communities had poor access to health services and became hotspots for post-disaster malaria and diarrheal disease outbreaks [33].

In South Africa, the eThekwini Municipality’s Kennedy Road informal settlement, situated in KwaZulu-Natal, has become a hotspot for climate-sensitive health risks. The 2023 floods overwhelmed pit latrines, mixing sewage with surface water. The local PHC mobile unit serving the area faced service interruptions due to impassable roads and power cuts affecting the refrigeration of vaccines and antibiotics [34]. Overcrowding and poor drainage also increased malaria transmission potential as stagnant water persisted, while cholera outbreaks exposed gaps in community-level health education and sanitation infrastructure.

Across these diverse settings, PHC systems are struggling to cope with the intensification of climate-sensitive diseases. Clinics often operate without early warning systems that integrate meteorological and epidemiological data. Community health workers lack training in climate–health links and resources to implement preventative measures such as chlorination or vector control. Supply chains for medicines and vaccines are highly vulnerable to weather-related disruptions, particularly in hard-to-reach rural areas [3,8].

Furthermore, PHC facilities in Nsanje, Buzi, Gwanda, and Kennedy Road all reflect systemic weaknesses: inadequate infrastructure, poor intersectoral coordination, and limited community engagement. For example, in Buzi, post-Idai reconstruction excluded health infrastructure in adaptation plans, while in Gwanda, siloed approaches to water and health policies have led to uncoordinated responses. These issues reflect broader governance and financing challenges that impede the resilience of health systems in climate-vulnerable areas.

**Table 4 ijerph-22-01242-t004:** Interplay between climatic shifts and disease patterns in Malawi, Zimbabwe, Mozambique, and South Africa.

Country	Disease	Recent Data and Trends	Climate Linkages	Sources
Malawi	Cholera	Between March 2022 and May 2023, Malawi experienced its deadliest cholera outbreak, with over 58,000 cases and more than 1700 deaths.	Extreme weather events, including cyclones, flooding, and damaged water infrastructure, led to widespread contamination and cholera outbreaks.	[35]
	Malaria remains a significant public health concern, with climate change threatening progress in control efforts.	Rising temperatures and altered rainfall patterns are expanding mosquito habitats, increasing malaria transmission.	Rising temperatures and altered rainfall patterns are expanding mosquito habitats, increasing malaria transmission.	[24]
South Africa	Malaria	Malaria cases are rising, particularly in Limpopo and Mpumalanga provinces, regions previously considered low-risk.	Climate change-induced flooding and warmer temperatures are creating favourable conditions for mosquito proliferation, expanding malaria transmission zones.	[36]
	Cholera	South Africa experienced cholera outbreaks in 2023–2024, with over 1300 suspected cases and 47 deaths reported.	Flooding and infrastructure failures have led to water contamination and cholera spread.	[31]
Mozambique	Cholera	Significant cholera outbreaks occurred following Cyclones Idai and Kenneth, with thousands of cases reported.	Cyclones and subsequent flooding led to water contamination and cholera outbreaks.	[37,38]
	Malaria	Mozambique continues to report high malaria cases, with climate change posing additional challenges to control efforts.	Increased rainfall and temperatures have expanded mosquito breeding sites, sustaining high malaria transmission rates.	[26]
Zimbabwe	Malaria	Malaria incidence has shown increasing trends, particularly in highland areas previously considered low-risk.	Climate change has made highland regions more suitable for malaria transmission due to increased temperatures and precipitation.	[38]
	Cholera	Zimbabwe has experienced recurrent cholera outbreaks, with significant cases reported in recent years.	Heavy rainfall and flooding have compromised sanitation systems, facilitating the spread of cholera.	[39]

Climate-sensitive diseases are proliferating under new environmental conditions across the SADC region. Warmer temperatures and erratic rainfall patterns have increased the geographic range, seasonal transmission, and intensity of vector-borne diseases such as malaria and dengue fever. For instance, elevated temperatures have expanded the breeding range of Anopheles mosquitoes into highland and peri-urban zones in Zimbabwe and Malawi, areas that were historically considered malaria-free [10]. Similarly, cholera, typhoid, and diarrheal diseases have surged following extreme flooding and prolonged droughts that compromise water infrastructure and sanitation systems [36].

Nsanje District in Malawi witnessed a severe cholera outbreak in 2022 after Cyclone Ana caused widespread flooding and water contamination, resulting in over 1200 deaths and overwhelming local Primary Health Care (PHC) clinics. In Gwanda, Zimbabwe, recurrent droughts have depleted clean water sources, increasing diarrheal diseases, particularly in children under five, who lack consistent access to oral rehydration therapy and sanitation [31]. Buzi District, Mozambique was heavily affected by Cyclone Idai in 2019 and hence experienced a threefold rise in cholera cases, while malaria became endemic due to stagnant floodwaters, as well as collapsed drainage systems [29]. Kennedy Road informal settlement in KwaZulu-Natal, South Africa, suffered a spike in waterborne diseases following the 2023 floods, which disrupted sanitation and exposed gaps in the province’s disease early warning and PHC outreach systems.

### 5.2. Climate-Driven Health Crises and the Gaps in Institutional Preparedness of Health Surveillance Systems in the SADC Region

The Southern African Development Community (SADC) region has not only witnessed an increasing frequency and severity of climate extremes, but also an equal proportion of vector and waterborne disease outbreaks, particularly cholera and malaria. However, these outbreaks are not solely a result of climatic extremes such as floods and droughts but also reveal deep symptomatic institutional and operational failures within SADC national health systems. Chief among these is the systemic inability to integrate climate intelligence into health surveillance, preparedness, and response mechanisms, leaving frontline Primary Health Care (PHC) systems reactive and chronically underprepared.

A significant barrier to early outbreak detection and rapid public health response in the SADC region is the lack of integration between meteorological data and public health surveillance systems. Despite the growing evidence base linking climate variability to disease incidence [27], most national disease surveillance systems continue to function in epidemiological isolation, with limited or no input from climate forecasting tools. For example, Malawi’s 2022 cholera outbreak in Nsanje District, intensified by post-Cyclone Ana flooding, could have been anticipated through early warning systems combining rainfall projections with water quality and sanitation risk mapping. Yet, the Ministry of Health’s disease surveillance did not include precipitation or hydrological indicators in its epidemic risk modelling, delaying preemptive public health interventions [28].

Similarly, in Buzi and Nhamatanda districts in Mozambique, the aftermath of Cyclone Idai in 2019 led to widespread cholera and malaria outbreaks. While rainfall anomalies were accurately forecast by the Mozambique National Institute of Meteorology (INAM), the health sector failed to activate its preparedness mechanisms due to the absence of institutional linkages between INAM and provincial health departments [29]. This institutional disjuncture not only delayed the deployment of vaccines and oral rehydration kits but also compromised vector control activities.

### 5.3. Operational Gaps in Primary Health Care Systems

The impact of climate-sensitive diseases on health outcomes is magnified by operational deficiencies within PHC systems, particularly in rural and informal settlements. This includes inadequate supply chain systems that fail during floods or infrastructure damage, hindering timely access to medications and diagnostic tools. In Chimanimani, Zimbabwe, malaria diagnostic kits and insecticide-treated nets were unavailable in several rural clinics following flooding in 2020, despite warnings from local disaster agencies. Another gap is associated with workforce shortages and training gaps, where frontline health workers lack skills to recognize and respond to climate-related disease patterns. In KwaZulu-Natal’s informal settlements, community health workers were unprepared to identify cholera symptoms during the 2023 outbreak, contributing to delayed referrals and higher case fatality rates. Lastly, there is lack of mobile or digital reporting infrastructure in remote areas, which delays the aggregation of outbreak data and weakens national situational awareness. In Phalombe District, Malawi, health centres lacked the digital infrastructure to report real-time cholera cases during the 2022 surge.

### 5.4. Poor Institutional Coordination and Crisis Governance

A recurring pattern across SADC countries is the fragmentation of roles and responsibilities between health ministries, disaster risk management authorities, meteorological services, and water utilities. These actors often operate in siloed institutional frameworks, undermining coordinated planning, resource pooling, and shared early warning protocols. For instance, in Manica Province, Mozambique, the absence of joint planning between the Provincial Health Directorate and the National Institute for Disaster Management (INGD) during seasonal flood events has led to duplicated efforts and missed opportunities for coordinated community-level interventions, such as pre-flood chlorination campaigns or temporary relocation of health services to safer ground. Likewise, Zimbabwe’s epidemic preparedness committees, which should coordinate multisectoral responses to climate-sensitive diseases, are often inactive at the district level or lack clear standard operating procedures [26]. This results in delays in the activation of emergency operations, late vaccine procurement, and reactive case management once outbreaks have already peaked.

### 5.5. Absence of Predictive, Localized Risk Modelling

While global and regional climate–health forecasting systems such as the WHO-WMO Climate and Health Outlooks and the SADC Climate Services Centre provide seasonal projections, these tools are rarely downscaled or operationalized at subnational levels in southern Africa. Health systems across the region continue to rely primarily on historical outbreak data, limiting their ability to engage in predictive risk modelling, conduct spatially targeted preparedness, or pre-position medical supplies in high-risk areas [1,2].

A recent study in Malawi found that although advanced rainfall forecasts were available for the Lake Chilwa Basin, they were not integrated into health sector early warning systems. This contributed to inadequate cholera preparedness during the 2022 outbreak, despite early signals of heightened flood risk [3]. In South Africa, the absence of microclimate and localized hazard modelling has undermined the identification of vulnerable urban settlements prone to heatwaves and flash floods, as demonstrated in post-event analyses of the 2022 and 2023 KwaZulu-Natal flood disasters [4,5]. This reflects a wider challenge in aligning climate services with public health response frameworks, particularly at municipal and district levels [6].

### 5.6. Underfunded Health Adaptation and Resilience Initiatives

Primary Health Care (PHC) systems across Southern Africa remain under-resourced to meet the growing burden of climate-related health impacts. National budgets typically lack earmarked funds for health-related climate adaptation, and there is limited integration of health priorities into climate finance mechanisms such as the Green Climate Fund or the Adaptation Fund [7]. This financial gap contributes to the chronic underinvestment in climate-resilient infrastructure, early warning systems, and community-based health preparedness [8,9].

For instance, while Mozambique’s National Adaptation Plan recognizes health as a climate-sensitive sector, its implementation is hampered by resource constraints, resulting in fragmented and donor-dependent interventions. A review by Mthembu and colleagues (2023) noted that the country’s health adaptation efforts lacked continuity and were overly reliant on short-term projects, impeding systemic strengthening [10]. Similar patterns are observed across the region, where national adaptation planning often does not translate into resilient and sustained PHC delivery in climate-stressed settings [11].

### 5.7. Direct Health Impacts of Rising Temperatures

Temperature-related mortality is a significant concern in South Africa, where 3.4% of deaths are attributed to cold or heat exposure, with the elderly, infants, and pregnant women being most vulnerable due to impaired thermoregulation and limited adaptive capacity [35]. A 1 °C temperature rise increases overall mortality by 1%, rising to 2% for those over 65 [40]. Similar trends are observed in Malawi and Mozambique, where heatwaves exacerbate maternal and child mortality risks in already strained health systems [41].

Occupational heat stress is escalating, particularly in mining, agriculture, and outdoor labour. Historical data from South African mines show that heat-related deaths rise nearly fivefold when temperatures exceed 34 °C [42]. In Upington South Africa, productivity is affected by sunburn, exhaustion, and productivity losses [40], while in Malawi and Zimbabwe, farm labourers face increased heat exposure with minimal protective measures. Urban heat islands amplify risks in informal settlements, where poorly insulated homes can be 4–5 °C hotter than outdoors. Replacing informal housing with insulated structures could halve heat-related deaths [42]. Schools and clinics also face dangerous indoor temperatures, with rural health facilities in South Africa exceeding outdoor heat levels by 4 °C, increasing risks for patients and staff [43].

Malaria transmission is shifting due to warming, particularly in South Africa’s Limpopo Province and neighbouring Mozambique, where temperature and rainfall increases trigger outbreaks [43,44]. Anopheles arabiensis, a key malaria vector, may have an expanded range in Southern Africa while declining elsewhere [43]. Dengue, Zika, and Rift Valley Fever (RVF) risks are rising with the spread of Aedes aegypti and Aedes albopictus mosquitoes [44]. RVF outbreaks in South Africa shift between Karoo and grassland regions during La Niña and El Niño cycles [45]. Similarly, cholera outbreaks in KwaZulu-Natal have been linked to warmer sea temperatures promoting pathogen-carrying copepods [46].

Diarrheal diseases surge with heat and flooding. In Cape Town, a 5 °C rise in minimum temperatures increased child diarrhea cases by 40% [47], a trend mirrored in Limpopo Province of South Africa [47]. Listeria outbreaks in South Africa are also climate-sensitive, with higher temperatures accelerating bacterial growth in food and water [47].

### 5.8. Mental Health and Social Consequences

Climate-related disasters compound South Africa’s high mental health burden, driven by poverty, violence, and HIV [48]. While research gaps persist, droughts have been linked to rising farmer suicides [48], and migration due to land degradation may increase gender-based violence and risky sexual behaviours [49].

Women and children bear the brunt of climate shocks. During droughts, women travel farther for water, while children face malnutrition and respiratory illnesses from dust and pollen [28]. HIV-positive individuals (8 million in South Africa alone) are at higher risk from heat-aggravated infections and disrupted health care [50]. Migration to urban slums heightens HIV exposure and xenophobic violence, particularly in South Africa [48,49].

### 5.9. Implications for Primary Health Care Systems

Primary Health Care (PHC), often defined as the first level of contact between individuals and the health system, is at the frontline of responding to climate-sensitive health threats. However, across the SADC region, PHC systems are under-resourced, unevenly distributed, and increasingly maladapted to evolving disease risks. Climate-induced health burdens have several implications for PHC systems:

Infrastructure vulnerability has been common. Flooding and cyclones have damaged health facilities, especially in rural Mozambique, Malawi, and eastern Zimbabwe, leading to service interruptions, loss of medical supplies, and displacement of health workers. Service delivery has also been constrained. PHC clinics in rural areas like Nsanje (Malawi) or Mutare South (Manicaland, Zimbabwe) often lack reliable access to clean water, electricity, or refrigeration essential for storing vaccines and reagents during outbreaks. The scale and frequency of climate-related disease outbreaks overwhelm already understaffed PHC facilities, especially community clinics that serve dispersed rural populations. Inadequate integration of climate data with disease surveillance delays detection and containment of outbreaks, as seen in delayed responses to cholera in Buzi and malaria surges in Zambezi Valley (Mozambique–Zimbabwe border). In places like KwaZulu-Natal, informal PHC responses (e.g., chlorination campaigns) are reactive and episodic rather than preventive and sustained, revealing the absence of climate-informed adaptation planning at the PHC level.

### 5.10. Systemic and Policy Challenges

Despite the formal inclusion of climate change in several national health adaptation strategies, significant implementation gaps still persist. While many National Adaptation Plans (NAPs) across the SADC region acknowledge the health–climate nexus, they rarely translate this recognition into localized and operational strategies within primary health care (PHC). Financing challenges exacerbate the problem: PHC budgets remain static or even decline, while external resources often target single diseases such as malaria or cholera, without integration into broader health system strengthening [31]. This fragmented approach is compounded by weak cross-sectoral coordination. The interlinkages between water, health, and agriculture are insufficiently addressed, resulting in siloed interventions. For instance, food insecurity during droughts heightens child malnutrition and vulnerability to infections, yet agricultural and health responses remain poorly aligned in PHC planning. The consequences of these systemic weaknesses are evident in the recurrence of climate-sensitive disease outbreaks. As shown in Table 5, cholera and malaria continue to resurface across several SADC countries, underscoring the urgent need for integrated, climate-resilient PHC strategies that bridge sectors and move beyond disease-specific interventions. 

The data for Zimbabwe’s Gwanda district is limited; however, reports indicate a significant increase in diarrheal diseases due to drought and water shortages. In Mozambique, precise case numbers for malaria post-Cyclone Idai are not specified, but the conditions were conducive to a surge in cases. The cholera outbreak in South Africa’s Hammanskraal was one of the most severe in recent history, highlighting systemic issues in water management. 

### 5.11. Climate-Induced Shifts in Disease Treatment as Burden and Strain on Frontline Primary Health Care in Vulnerable Southern African Settings Increases

Climate change is driving a profound shift in the epidemiological landscape across Southern Africa, disproportionately affecting vulnerable rural communities and informal urban settlements. In regions with fragile infrastructure and low adaptive capacity, rising temperatures, erratic rainfall, and extreme weather events are fuelling outbreaks of waterborne and vector-borne diseases, placing unprecedented strain on Primary Health Care (PHC) systems that are often under-resourced and underprepared [1,9].

Climate change is profoundly altering the landscape of disease burden in the Southern African Development Community (SADC) region, with increasing evidence of its role in driving the frequency and intensity of vector-borne and waterborne disease outbreaks. This shifting epidemiological terrain presents complex challenges for already fragile Primary Health Care (PHC) systems. A critical exploration of institutional, socio-cultural, and operational factors reveals the interplay between climate-induced stressors and systemic health vulnerabilities, particularly in rural and peri-urban communities across Malawi, Mozambique, Zimbabwe, and South Africa.

The theoretical basis for understanding these dynamics is grounded in McMichael’s eco-social model, which emphasizes the intertwined relationship between ecological disruption and social determinants in shaping public health outcomes. This approach is particularly salient in SADC, where climate events such as cyclones, floods, and droughts interact with poverty, limited infrastructure, and institutional fragility to produce compounded risks. Complementing this is the One Health paradigm, which underscores the integrated health of people, animals, and the environment—an essential framework in a region where zoonotic spillovers and vector ecology are sensitive to environmental changes. The Health System Resilience Framework further deepens this analysis by identifying the capacities required for health systems to absorb shocks, adapt processes, and transform in the face of ongoing climate threats.

In Malawi, the cholera outbreak that surged after Cyclone Ana in 2022 exemplifies the failure of systems to bridge climate intelligence with public health readiness. While meteorological agencies had warned of extreme rainfall and flood risk, this information was not effectively translated into health preparedness actions. The resulting contamination of water sources and displacement of vulnerable populations created a fertile environment for cholera transmission, particularly in districts like Nsanje and Chikwawa. The Ministry of Health’s delayed deployment of resources and limited availability of oral rehydration salts and clean water infrastructure significantly undermined the PHC system’s capacity to manage the crisis.

Mozambique’s central provinces of Sofala and Manica continue to face recurrent challenges linked to cyclonic activity and stagnant water, which fuel both cholera and malaria outbreaks. Following Cyclone Idai in 2019 and Cyclone Freddy in 2023, health infrastructure in Buzi and Nhamatanda districts was extensively damaged, leaving communities without access to care. Despite efforts by NGOs and international agencies to rebuild clinics and supply essential medicines, the disconnect between disaster risk reduction strategies and routine health planning remains stark. Health surveillance continues to operate in silos, with minimal integration of environmental monitoring tools such as GIS to predict vector-breeding zones or flood-prone areas.

In Zimbabwe, malaria remains a persistent threat in rural parts of Matabeleland North and East, particularly in Binga, Hwange, and Tsholotsho districts. Seasonal rains, coupled with poorly drained infrastructure and limited vector control, contribute to periodic malaria surges. Although the country has made significant gains in expanding insecticide-treated net coverage, inconsistencies in distribution and lack of community health worker training in climate-sensitive surveillance reduce the effectiveness of interventions. The 2021 data indicating over 265 malaria-related deaths illustrates the failure to translate climatic risk forecasts into targeted PHC strategies.

South Africa’s experience, particularly in informal settlements across KwaZulu-Natal such as eThekwini (Durban) and Hammarsdale, has highlighted the vulnerability of peri-urban populations to climate-linked health threats. In 2022 and 2023, severe flooding led to the contamination of water systems, prompting spikes in diarrheal diseases. The Hammanskraal cholera outbreak of 2023 revealed a stark breakdown in interdepartmental coordination, with misaligned responsibilities between water, health, and municipal authorities delaying the containment response. Community engagement was insufficient, and health facilities were overwhelmed, underscoring a systemic failure to anticipate and prepare for the health impacts of climate events.

Across these settings, a recurrent theme is the fragmented nature of surveillance systems, which fail to leverage available climate data to inform outbreak forecasting and PHC readiness. Institutional inertia limited political will, and under-resourced public health departments have hindered the adoption of integrated early warning systems. This is compounded by social and cultural factors that impede timely care-seeking. In many rural communities, traditional healers remain the first point of contact, and beliefs about disease causation are shaped more by cultural narratives than biomedical explanations. Language barriers, mistrust in formal health institutions, and gender dynamics further complicate efforts to implement public health campaigns.

The operational capacity of PHC systems in the region remains constrained by inadequate infrastructure, a shortage of trained personnel, and logistical bottlenecks in emergency response. Facilities lack basic infection control supplies, water and sanitation services, and reliable cold chains for vaccine storage. These deficiencies are most acute in rural and remote areas, where communities face long travel distances and inconsistent access to care. The challenges are not only structural, but also epistemological. Climate change is still perceived as an environmental issue rather than a health emergency, leading to limited mainstreaming of climate adaptation into health planning and budgeting processes.

### 5.12. Encroachment of Diseases in Regions Usually Devoid of Such Diseases Due to Climate Change

Climate change is catalyzing the geographic redistribution of vector-borne and waterborne diseases in the SADC region, expanding their reach into areas that have historically been less affected or even considered non-endemic. This spatial shift is driven by rising temperatures, erratic rainfall, flooding, and increased humidity conditions that create new ecological niches for disease vectors and pathogens. The encroachment of these diseases into previously unaffected areas poses a serious challenge to health systems that are often unprepared, both in infrastructure and disease surveillance capacity.

One prominent example is the changing distribution of malaria in Zimbabwe. Traditionally, malaria transmission was largely confined to low-lying, humid regions such as the Zambezi Valley and parts of the northeastern border with Mozambique. However, recent evidence shows an upward shift in malaria cases into previously malaria-free highland areas such as Nyanga in Manicaland and parts of Matabeleland South, including Gwanda and Insiza districts. These shifts have been linked to changing rainfall patterns and rising minimum temperatures, which extend the breeding season of Anopheles mosquitoes and reduce the extrinsic incubation period of Plasmodium falciparum [27].

In Malawi, cholera outbreaks, once typically confined to lakeshore communities and flood-prone southern districts like Nsanje and Chikwawa, have now emerged in highland and peri-urban areas, including Mzimba and Lilongwe peri-urban settlements. The 2022–2023 cholera epidemic, Malawi’s worst in decades, recorded cases in nearly all districts, with over 56,000 cases and more than 1600 deaths by mid-2023 [28]. The expansion of cholera into these areas is attributed to extreme weather events, including Cyclones Ana and Freddy, which damaged water infrastructure and forced displacement, increasing exposure to unsafe water and poor sanitation.

Mozambique’s central provinces of Sofala and Manica have long been vulnerable to waterborne diseases due to recurring cyclones, but recent outbreaks have extended inland, affecting upland districts like Gondola and Barue that had rarely recorded cholera historically. This inland spread is linked to infrastructure collapse, floodwater contamination, and poor post-disaster recovery planning, revealing significant operational weaknesses in disease containment systems [29]. South Africa has also witnessed the re-emergence and spread of diseases typically considered rare or limited to specific zones. The Hammanskraal cholera outbreak in 2023 marked a significant public health event in a region not historically known for cholera transmission. The contamination of drinking water systems due to municipal failures and increased stormwater runoff created a conducive environment for Vibrio cholerae proliferation [53]. Concurrently, parts of KwaZulu-Natal, such as Umlazi and eThekwini, have reported an uptick in vector-borne diseases like dengue and chikungunya, which were previously uncommon. These cases are likely linked to increased urban flooding, waste accumulation, and warmer temperatures, which promote mosquito breeding in urban slums and informal settlements.

These epidemiological shifts underscore the urgent need for integrated climate–health surveillance systems that combine environmental monitoring (e.g., temperature, rainfall, humidity) with real-time disease reporting. The current health surveillance systems remain reactive and disconnected from meteorological and ecological data, leading to delayed responses when diseases emerge in new areas. Moreover, PHC systems in these newly affected areas often lack the diagnostic capacity, vector control programs, and community awareness necessary to contain outbreaks swiftly.

The encroachment of diseases into new geographies also exposes deep socio-political and economic inequities. Poor, marginalized communities often residing in informal settlements or remote rural areas lack access to clean water, adequate sanitation, and reliable health services. In areas like Umlazi (KwaZulu-Natal), Binga (Matabeleland North), and peri-urban Lilongwe, climate-induced disease shifts exacerbate existing health disparities, straining overburdened health workers and exposing governance gaps in multisectoral coordination.

To respond effectively to these converging crises, it is imperative to foster systems that are not only reactive but also anticipatory. This requires embedding climate intelligence within health information systems, strengthening health worker training on climate-related health risks, and ensuring community voices shape intervention design. An integrated approach, anchored in equity, sustainability, and local knowledge, is critical for transforming PHC systems into climate-resilient institutions capable of protecting vulnerable populations in a warming world. In summary, climate change is not only intensifying existing disease burdens but also spatially transforming the public health landscape in the SADC region. This underscores the critical importance of proactive, adaptive health systems that can anticipate and respond to emerging threats, particularly in under-resourced and newly vulnerable communities.

## 6. Conclusions and Recommendations

The findings of this study underscore the increasingly complex and dynamic interface between climate change and public health in the SADC region, where climate-induced shocks, such as floods, droughts, and cyclones, are intensifying the burden of vector-borne and waterborne diseases. The spatial redistribution of diseases like cholera and malaria into previously unaffected areas reveals significant gaps in disease surveillance, health infrastructure, and institutional preparedness. These shifts are particularly acute in socioeconomically marginalized communities across Malawi, Mozambique, Zimbabwe, and South Africa, where frontline Primary Health Care (PHC) systems are often the first and sometimes only line of defence.

This study reveals that many of the institutional and operational weaknesses stem from the failure to integrate climate intelligence into health systems, fragmented disaster preparedness frameworks, and a lack of community-centred adaptation strategies. The challenges faced in areas like Gwanda (Zimbabwe), Gondola (Mozambique), Hammanskraal (South Africa), and Chikwawa (Malawi) demonstrate the urgent need for transformative, multisectoral action that strengthens health system resilience from the ground up.

Based on the comparative analysis and critical insights from the selected case studies, the following recommendations are proposed to foster climate-resilient PHC systems across the SADC region:

First, there is a need for the integration of climate intelligence into surveillance systems. This establishes interoperable platforms that combine meteorological data with epidemiological surveillance, enabling predictive analytics and early warning systems to be put in place. Real-time monitoring must be localized to high-risk zones to facilitate anticipatory interventions during flood and drought cycles.

Second, there is also a need to climate-proof PHC infrastructure. This involves reinforcing the structural integrity of PHC clinics that are located in disaster-prone areas such as Nsanje (Malawi), Beira (Mozambique), and Umlazi (South Africa). Thus, PHC facilities must be elevated above flood lines and secured by the installation of off-grid renewable energy systems, integrated rainwater harvesting facilities, and on-site water purification technologies.

Thirdly, human resource capacity needs strengthening by designing and implementing training programs for PHC personnel that focus on climate-sensitive diseases, outbreak management, and psychosocial support during climate disasters. Districts like Manica in Mozambique and Matabeleland South in Zimbabwe would benefit from mobile training units and regional knowledge-sharing platforms.

Empowering community health workers and traditional leaders to co-produce localized adaptation strategies that integrate Indigenous Knowledge Systems for disease forecasting and health communication is the fourth recommendation. This approach should reflect cultural norms and local epidemiological realities, ensuring trust and uptake in regions such as Binga (Zimbabwe) and Chikwawa (Malawi).

Other recommendations include multisectoral and decentralized governance, investing in GIS and digital mapping for risk analysis, and mobilizing regional and international funding. GIS is particularly helpful since it helps to map disease hotspots, identify infrastructure gaps, and track climate-related hazards over time. These maps are used for planning health resource allocation, emergency preparedness, and guiding outreach in underserved communities.

Thus, climate change is not merely an environmental issue but is a profound public health challenge that demands proactive, context-specific, and equity-driven solutions. Strengthening PHC systems in the SADC region will require not only technical adaptation but also political will, intersectoral collaboration, and deep engagement with the communities most affected. These recommendations offer a roadmap for navigating this urgent and evolving landscape.

## Figures and Tables

**Table 1 ijerph-22-01242-t001:** Climate change indicators in four SADC countries (2015–2023).

Indicator	Malawi	Zimbabwe	Mozambique	South Africa	Source
Average Temperature Increase (°C)	+1.2	+1.4	+1.1	+1.3	[17,18]
Rainfall Variability (mm/yr)	−15%	−20%	+10% (cyclones)	−12%	[18,19,20]
Type of Extreme Weather Events	Floods	Droughts	Cyclones/Floods	Heatwaves/Floods	[21,22]

**Table 2 ijerph-22-01242-t002:** Infectious disease burden linked to climate change.

Disease	Malawi	Zimbabwe	Mozambique	South Africa	Source
Malaria Cases (/100 k)	320 → 380 (+19%)	280 → 340 (+21%)	410 → 500 (+22%)	50 → 65 (+30%)	[23]
Cholera Outbreak index	12	15	25	8	[24]
Dengue Fever	Low	Moderate	High (post-cyclone)	Low	[22]

**Table 3 ijerph-22-01242-t003:** Impact on Primary Health Care (PHC) systems.

Phenomenon Index	Malawi	Zimbabwe	Mozambique	South Africa	Source
Clinic Overcrowding (Post-Disaster)	+40%	+35%	+60%	+25%	[25]
Drug Stockouts (%)	18%	22%	30%	12%	[22]
Health Worker Shortage (/10 k)	2.1	1.8	2.5	3.0	[24]

**Table 5 ijerph-22-01242-t005:** Cholera and malaria outbreaks in selected SADC countries.

Country	District/Province	Disease	Period	Cases	Deaths	Contributing Factors	References
Malawi	Nsanje	Cholera	Mar–Apr 2022	76	4	Post-Cyclone Ana flooding, contaminated water sources, inadequate sanitation facilities	[24]
	Nationwide (29 districts)	Cholera	Mar–Nov 2022	7626	219	Widespread lack of clean water and sanitation, overwhelmed health systems	[22]
	Blantyre and Lilongwe	Cholera	Dec 2022–Jan 2023	2773	137	Urban overcrowding, poor waste management, and reliance on contaminated water sources	[51]
Mozambique	Buzi and Nhamatanda (Sofala)	Cholera	Mar–Apr 2019	1000	N/A	Aftermath of Cyclone Idai, destruction of infrastructure, and limited access to clean water	[52]
	Buzi and Nhamatanda (Sofala)	Malaria	Mar–Apr 2019	N/A	N/A	Stagnant floodwaters creating breeding grounds for mosquitoes	[52]
Zimbabwe	Gwanda (Matabeleland South)	Diarrheal Diseases	2020	N/A	N/A	Prolonged drought, water scarcity, inadequate sanitation infrastructure	[29]
	Mutasa, Chipinge, Buhera (Manicaland)	Malaria	Jan–Jun 2021	226,752	266	Higher rainfall, warmer climate, lack of indoor residual spraying in time	[23]
South Africa	Hammanskraal (Gauteng)	Cholera	May-Jun 2023	149	47	Contaminated water supply, delayed response, inadequate infrastructure	

(Source: Compiled by authors).

## Data Availability

The original contributions presented in this study are included in the article Further inquiries can be directed to the corresponding author(s).

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
