# Peer review of "Health Inequalities in Primary Care: A Comparative Analysis of Climate Change-Induced Expansion of Waterborne and Vector-Borne Diseases in the SADC Region"

_ijerph, 2025, doi:10.3390/ijerph22081242_

Round 1

Reviewer 1 Report

Comments and Suggestions for Authors

Congratulations on this important and timely study. Your work offers valuable insights into the impacts of climate change on health and primary care systems in the SADC region. I particularly appreciate the regional perspective and the integration of case-based analysis. That said, I have identified a few areas where the manuscript’s presentation could be improved, which I outline below as constructive suggestions.

First, the methodology section would benefit from further elaboration. Currently, it lacks sufficient detail on how the study was conducted. In particular, the processes of quantitative data collection and analysis are not clearly described. This section should be expanded to a level of clarity that would allow another researcher to replicate the study.

Second, to enhance structure and readability, a separate section heading should be added for the Findings/Results section.

Additionally, a few editorial issues need to be addressed:

  • In line 194, the phrase “table below” should be replaced with the appropriate table number.

  • In Table 2, the label “Table xx” appears to be a placeholder and should be updated.

  • In line 260, “Districtin” should be corrected to “District in.”

Finally, while the case-based analysis is compelling, the manuscript does not present country-level quantitative findings, even though epidemiological and meteorological data are referenced in the methodology. If such data will not be included in the results, I recommend revising the methodology section to better reflect the qualitative scope of the study and ensure alignment between methods and findings.

In conclusion, I believe this manuscript has strong potential to contribute meaningfully to the climate and health literature. With some refinements in methodological transparency, structural clarity, and editorial accuracy, the paper will be significantly strengthened.

Author Response

Comment 1:The methodology section would benefit from further elaboration. Currently, it lacks sufficient detail on how the study was conducted. In particular, the processes of quantitative data collection and analysis are not clearly described. This section should be expanded to a level of clarity that would allow another researcher to replicate the study.

Response:

Comment 2: to enhance structure and readability, a separate section heading should be added for the Findings/Results section.

Response: 

editorial issues need to be addressed:

  • In line 194, the phrase “table below” should be replaced with the appropriate table number.

  • In Table 2, the label “Table xx” appears to be a placeholder and should be updated.

  • In line 260, “Districtin” should be corrected to “District in.”

Response: We have attached a table with responses and also  an article with rack changes

Reviewer 2 Report

Comments and Suggestions for Authors

This is a very interesting topic and one that is very relevant given the climate emergency in countries on the African sub continent. I do have some concerns about the way the research was conducted however.

You mention using quantitative data however there is no quantitative data presented in the paper. The results section reads more like a literature review. The tables are basic and do not present the results in the way that the methods are described. I think the entire methodology needs to be thought through and linked to the results in a much more cohesive way.

The reference list is incomplete. There are many references in the text that are not included in the reference list. This must be rectified. (e.g. McMichael, 2020; Wyndham, 1965 and many more).

Author Response

Comment:You mention using quantitative data however there is no quantitative data presented in the paper.

Response:

Comment: The results section reads more like a literature review. The tables are basic and do not present the results in the way that the methods are described. I think the entire methodology needs to be thought through and linked to the results in a much more cohesive way.

Response:

Comment: The reference list is incomplete. There are many references in the text that are not included in the reference list. This must be rectified. (e.g. McMichael, 2020; Wyndham, 1965 and many more).

Response: We agree with the comment. We have reviewed our reference list, updated it and worked on the missing references. This has been done.
